# Chemoprevention and Lifestyle Modifications for Risk Reduction in Sporadic and Hereditary Breast Cancer

**DOI:** 10.3390/healthcare11162360

**Published:** 2023-08-21

**Authors:** Eliza Del Fiol Manna, Davide Serrano, Gaetano Aurilio, Bernardo Bonanni, Matteo Lazzeroni

**Affiliations:** Division of Cancer Prevention and Genetics, IEO European Institute of Oncology IRCCS, 20141 Milan, Italy; davide.serrano@ieo.it (D.S.); gaetano.aurilio@ieo.it (G.A.); bernardo.bonanni@ieo.it (B.B.); matteo.lazzeroni@ieo.it (M.L.)

**Keywords:** chemoprevention, lifestyle, risk reduction, sporadic breast cancer, hereditary breast cancer

## Abstract

Female breast cancer is the most commonly diagnosed malignancy worldwide. Risk assessment helps to identify women at increased risk of breast cancer and allows the adoption of a comprehensive approach to reducing breast cancer incidence through personalized interventions, including lifestyle modification, chemoprevention, intensified surveillance with breast imaging, genetic counseling, and testing. Primary prevention means acting on modifiable risk factors to reduce breast cancer occurrence. Chemoprevention with tamoxifen, raloxifene, anastrozole, and exemestane has already shown benefits in decreasing breast cancer incidence in women at an increased risk for breast cancer. For healthy women carrying BRCA 1 or BRCA 2 pathogenic/likely pathogenic (P/LP) germline variants, the efficacy of chemoprevention is still controversial. Adopting chemoprevention strategies and the choice among agents should depend on the safety profile and risk–benefit ratio. Unfortunately, the uptake of these agents has been low. Lifestyle modifications can reduce breast cancer incidence, and the recommendations for BRCA 1 or BRCA 2 P/LP germline variant carriers are comparable to the general population. This review summarizes the most recent evidence regarding the efficacy of chemoprevention and lifestyle interventions in women with sporadic and hereditary breast cancer.

## 1. Introduction

Female breast cancer is the most commonly diagnosed malignancy worldwide [1]. GLOBOCAN 2020 estimated 2.3 million new cases and 684,996 deaths from breast cancer and showed that female breast cancer surpassed lung cancer as the most commonly diagnosed cancer [1]. One in eight women (12.9%) will develop breast cancer in their lifetime [2,3]. In addition, the burden of breast cancer is rising worldwide in premenopausal and postmenopausal women [1,4].

Familial breast cancer accounts for 15% to 20% of all breast cancer cases, and about 5% to 10% of breast cancers are due to genetic predisposition [5,6,7]. Hormonal and reproductive factors, mammographic breast density, and proliferative breast disease explain approximately half of breast cancer cases [8].

In 2010, the fraction of breast cancer attributable to lifestyle and environmental factors in the United Kingdom was around 27%, of which 18.5% was related to alcohol, diet, overweight, and obesity [9]. Primary prevention may represent an opportunity to act on modifiable factors and intervene before breast cancer develops. Therefore, preventive strategies focused on decreasing excess body weight, alcohol consumption, and increasing physical activity may impact the burden of breast cancer worldwide [1].

Individualized breast cancer risk assessment helps to identify women at increased risk of breast cancer, allowing them to benefit from personalized risk management strategies [10]. A comprehensive approach to reducing breast cancer incidence encompasses adopting personalized risk-reduction interventions, including lifestyle modification, chemoprevention, intensified surveillance with breast imaging, genetic counseling, and testing [11]. Women with an inherited P/LP germline variant, which confers a high risk for breast cancer, may benefit from risk reduction surgery, like bilateral salpingo-oophorectomy and risk-reducing mastectomy [12,13].

Evidence-based risk reduction strategies according to different risk categories should be implemented to decrease breast cancer burden. This review summarizes the most recent evidence regarding primary prevention, focusing on the efficacy of lifestyle interventions and chemoprevention according to breast cancer risk.

## 2. Chemoprevention

### 2.1. Chemoprevention for Sporadic Breast Cancer

Chemoprevention with tamoxifen or raloxifene (selective estrogen receptor modulators, SERMs) and anastrozole or exemestane (aromatase inhibitors, AIs) has shown to reduce breast cancer occurrence in women at increased risk of developing breast cancer [14,15,16,17,18]. The choice of the ideal agent should consider patient-specific risk factors (age, baseline comorbidities) and the adverse events of the different agents [3]. The ASCO clinical practice guidelines recommend the use of endocrine therapy with anastrozole (1 mg/d), exemestane (25 mg/day), raloxifene (60 mg/day), or tamoxifen (20 mg/day) for postmenopausal women with an increased risk of developing breast cancer [19]. Risk reduction agents are recommended only for individuals ≥ 35 years old because the utility of these agents in younger women is unknown [13]. For women aged ≥ 35 years who have completed childbearing, tamoxifen is still the standard of care [19,20]. Tamoxifen is the most studied agent and the only one indicated for premenopausal women, while all four agents may be prescribed for postmenopausal women [13]. 

Women who could benefit most from chemoprevention with endocrine therapy are those who “have been diagnosed with atypical (ductal or lobular) hyperplasia or lobular carcinoma in situ (LCIS) or have an estimated 5-year risk (according to the National Cancer Institute Breast Cancer Risk Assessment Tool) of at least 3%, a 10-year risk (according to the International Breast Intervention Study [IBIS]/Tyrer-Cuzick Risk Calculator) or have at least 5%, or a relative risk of at least four times the population risk for their age group if they are age 40 to 44 years or at least two times the population risk for their age groups if they are age 45 to 69 years” [19]. 

In 1992, the National Surgical Adjuvant Breast and Bowel Project initiated the Breast Cancer Prevention Trial (P-1 Study) following the observation that using tamoxifen for adjuvant therapy reduced the incidence of contralateral breast cancer [21]. The study randomized 13,388 women at increased risk for breast cancer to receive a placebo (n = 6707) or 20 mg/day tamoxifen (n = 6681) for five years. Increased risk was defined by age ≥ 60, or between 35 and 59, with a Gail model 5-year score > 1.66% or a previous history of lobular carcinoma in situ. After a median follow-up of 54.6 months, the results showed that tamoxifen reduced the risk of invasive breast cancer by 49% (two-sided *p* < 0.00001) and noninvasive breast cancer by 50% (two-sided *p* < 0.002). The incidence of estrogen receptor-positive (ER+) tumors was reduced by 69%, but no difference was observed in the occurrence of estrogen receptor-negative (ER-) tumors [21]. Tamoxifen administration increased the rates of endometrial cancer (RR 2.53; 95% CI 1.35–4.97) and pulmonary embolism in women aged ≥ 50 years (RR 3.19; 95% CI 1.12–11.15) [21]. After seven years of follow-up, the benefit remained for both invasive (RR 0.57; 95% CI 0.46–0.70) and noninvasive breast cancer (RR 0.63; 95% CI 0.45–0.89) [22].

Along this line, the first International Breast Cancer Intervention Study (IBIS-I) reported the risk reduction of invasive breast cancer with tamoxifen use as well [23]. This study randomized 7254 patients between 35 and 70 years old with a high risk for breast cancer to receive tamoxifen or a placebo for five years [23]. Increased risk for breast cancer was defined by age, family history, high-risk histology, or an estimated 10-year risk higher than 5%. Tamoxifen decreased breast cancer occurrence by 32% (95% CI 8–50, *p* = 0.013) [23]. In addition, at a median follow-up of 8 years, tamoxifen use reduced the incidence of all types of invasive breast cancer (RR 0.73; 95% CI 0.58–9.91, *p* = 0.004) [24].

The Royal Marsden Hospital study was a pilot randomized placebo-controlled trial that included healthy women with an increased risk of developing breast cancer based on strong family history (between October 1986 and June 1993). The study aimed to evaluate the efficacy of tamoxifen 20 mg/day for up to 8 years in reducing breast cancer incidence [25,26]. This study allowed women to continue or initiate hormone replacement therapy (HRT). No difference in breast cancer incidence between the groups was observed at 20-year follow-up (HR 0.78; 95% CI 0.58–1.04; *p* = 0.10). Of note, the incidence of ER+ tumors was significantly lower in the tamoxifen arm (HR 0.61; 95% CI 0.43–0.86; *p* = 0.005) [27].

The Italian Tamoxifen Prevention Study randomized 5408 healthy women—between 35 and 70 years old—who had undergone a previous hysterectomy to receive tamoxifen or placebo for five years. No difference in breast cancer incidence was observed in the overall study population at a median follow-up of 46, 81.2, and 109.2 months, respectively [28,29,30]. In the study population, however, only 13% (n = 702) of women could be considered at increased risk for breast cancer based on reproductive and hormonal characteristics. At 11 years of follow-up, in the subgroup analysis for the higher-risk women, the breast cancer rates were statistically reduced by tamoxifen (RR 0.24; 95% CI 0.1–0.59) [29].

Raloxifene is a second-generation SERM with similar anti-estrogenic effects and less endometrial stimulation than tamoxifen [13]. The placebo-controlled randomized Multiple Outcomes of Raloxifene Evaluation (MORE) trial evaluated the efficacy of raloxifene in reducing the risk of fracture in postmenopausal women with osteoporosis [31]. The authors randomized 7705 postmenopausal patients between 31 and 80 years to receive a placebo, or raloxifene 60 mg/day, or raloxifene 120 mg/day for three years. Raloxifene decreased the risk of vertebral fractures and increased bone mineral density in the femoral neck and spine [31]. After a median follow-up of 40 months, the relative risk of developing invasive breast cancer was 0.24 (95% CI 0.13–0.44). Raloxifene reduced only the incidence of ER+ breast cancer (RR 0.1; 95% CI 0.04–0.24). However, raloxifene increased the incidence of deep venous thromboses and pulmonary emboli but not the risk of endometrial cancer [31].

The Continuing Outcomes Relevant to Evista (CORE) trial evaluated the impact of 4 additional years of raloxifene on the incidence of invasive breast cancer in 4011 women enrolled in the MORE trial [32]. Raloxifene reduced the 4-year incidence of invasive breast cancer by 59% (HR 0.41; 95% CI 0.24–0.71) and invasive ER+ breast cancer by 61% (HR 0.34; 95% CI 0.18–0.66), without impacting the occurrence of ER- tumors [32]. In addition, raloxifene did not increase the risk of endometrial events or thromboembolism (RR 2.17; 95% CI 0.83–5.70) [32].

The Raloxifene Use for The Heart (RUTH) trial randomized postmenopausal women with a high risk for coronary heart disease to receive raloxifene or placebo [33,34]. According to the Gail model, 40% of the study participants had an increased risk for breast cancer. After a median follow-up of 5.6 years, raloxifene reduced the incidence of invasive breast cancer by 44% (95% CI 0.38–0.83) and decreased the occurrence of ER+ tumors by 55% (95% CI 0.28–0.72) but did not decrease the risk of noninvasive breast cancer and cardiovascular events [34].

The NSABP STAR trial (P-2 Study) compared the efficacy of tamoxifen versus raloxifene to reduce breast cancer incidence. A total of 19,747 postmenopausal women aged > 35 years with high risk for invasive breast cancer, based on the modified Gail model or with a personal history of LCIS, were randomized to receive tamoxifen 20 mg/day or raloxifene 60 mg/day for five years. The efficacy was similar (RR 1.02; 95% CI 0.82–1.28); however, thromboembolic events and cataracts occurred less frequently in the raloxifene group (RR 0.70; 95% CI 0.54–0.91 and RR 0.79; 95% CI 0.68–0.92, respectively) [35].

Other SERMs have shown a reduction in the incidence of invasive breast cancer in postmenopausal women with osteoporosis. Arzoxifene [36] reduced invasive breast cancer incidence by 56% (95% CI 0.26–0.76, *p*< 0.001) and lasofoxifene [37] by 79% (95% CI 0.08–0.55).

A meta-analysis with individual participant data from nine prevention trials assessed the efficacy of chemoprevention with four SERMs (tamoxifen, raloxifene, arzoxifene, and lasofoxifene) in reducing all breast cancers’ incidence during ten years of follow-up. The analysis evaluated data from 83,399 women during a median follow-up of 65 months. Breast cancer incidence decreased by 38% (HR 0.62; 95% CI 0.56–0.69), while the frequency of thromboembolic events increased with all SERMS (OR 1.73; 95% CI. 1.47–2.05, *p*<0.0001), and vertebral fractures reduced by 34% (HR 0.66, 95% CI 0.59–0.73) [14].

Aromatase inhibitors have also been evaluated for primary prevention in women with an increased risk for breast cancer. The MAP.3 randomized, placebo-controlled, double-blind trial randomized 4560 postmenopausal women ≥ 35 years old with moderately increased risk for breast cancer to receive a placebo or exemestane. Women eligible for the study were ≥60 years old, had a Gail 5-year score > 1.66%, had prior atypical ductal or lobular hyperplasia or lobular carcinoma in situ, or had a history of ductal carcinoma in situ with mastectomy. During a median follow-up period of 3 years, the annual incidence of invasive breast cancer decreased in patients receiving exemestane compared with placebo (HR 0.35; 95% CI 0.18–0.70, *p* = 0.002). The frequency of skeletal fractures, cardiovascular events, or deaths related to treatment were similar [17].

The international, double-blind, randomized, placebo-controlled IBIS-II trial assessed the efficacy and safety of anastrozole for preventing breast cancer in 3864 postmenopausal women at increased risk [38]. The study randomized 1920 women to receive anastrozole 1 mg/day and 1944 a placebo for five years. After a median follow-up of 5 years, anastrozole use decreased the incidence of breast cancer (HR 0.47; 95%CI 0.32–0.68, *p* < 0.0001); the reduction occurred mainly in high-grade tumors compared with intermediate- or low-grade tumors [38]. The use of anastrozole was associated with a 54% reduction in invasive ER+ breast cancer (HR 0.46; 95% CI 0.33–0.65, *p* < 0.0001) and a 59% decrease in ductal carcinoma in situ (HR 0.41; 95% CI 0.22–0.79, *p* = 0.0081), mainly in participants with ER+ tumors (HR 0.22; 95% CI 0.78–0.65, *p* < 0.0001). No significant difference was observed in overall deaths (HR 0.96; 95% CI 0.69–1.34, *p* = 0.82) or deaths for breast cancer. In addition, breast cancer incidence showed a significant continuing reduction in long-term follow-up [18].

A meta-analysis of six studies evaluated the efficacy and acceptability of breast cancer prevention agents in 50,927 women at above-average risk of developing breast cancer. Tamoxifen use reduced breast cancer risk compared to placebo (HR 0.68; 95% CI 0.62–0.76) but increased the risk of severe toxicity (RR 1.28; 95% CI 1.12–1.47), particularly endometrial cancer and thromboembolism. Aromatase inhibitor use reduced the risk of breast cancer by 53% (RR 0.47; 95% CI 0.35–0.63) but increased the risk of severe toxicity by 18% (RR 1.18; 95% CI 1.09–1.28), especially hot flashes, diarrhea, and arthralgia [15].

The US Preventive Services Task Force conducted a systematic review (46 studies) to evaluate medication use for the risk reduction of primary breast cancer in women [16]. In placebo-controlled trials, tamoxifen (RR 0.69; 95% CI 0.59–0.84), raloxifene (RR 0.44; 95% CI 0.24–0.80), exemestane, and anastrozole (RR 0.45; 95% CI 0.26–0.70) decreased the incidence of invasive breast cancer but did not reduce breast cancer-specific and all-cause mortality [16]. 

Tamoxifen, raloxifene, and aromatase inhibitors were associated with acute, long, and late adverse effects that differed between medications. Raloxifene was associated with a reduced incidence of vertebral fractures compared with tamoxifen (RR 0.61; 95% CI 0.53–0.73). Thromboembolic events occurred more frequently in patients receiving tamoxifen (RR 1.93: CI 95% 1.33–2.68) and raloxifene (RR 1.56: CI 95% 1.11–2.60) compared with placebo. In addition, tamoxifen increased the risk of endometrial cancer (RR 2.25; 95% CI 1.17–4.41) and cataracts (RR1.22: CI 95% 1.08–1.48) compared to the placebo. Vasomotor and musculoskeletal events varied by medication [16]. 

Concerns about the burden of adverse effects of chemoprevention raised doubts regarding the use of chemoprevention, considering the benefit overestimated, especially for healthy women [39]. In addition, the fear of side effects is a significant reason for the poor adherence to chemoprevention for breast cancer risk reduction [40,41]. However, the success of preventive therapy in reducing breast cancer incidence depends on adherence to therapy and the adequate uptake of chemopreventive agents. 

A systematic review including 24 articles with 21,423 women reported a pooled uptake of 16.3% (95% CI 13.6–19.0) of breast cancer prevention agents. In addition, the uptake of preventive agents was significantly higher in patients treated in trials (25.2%; 95% CI 18.3–32.2) than in routine care (8.7%; 95% CI 6.8–10.9, *p* < 0.001) [42]. 

A study evaluated whether chemoprevention uptake differs among women according to the presence of risk factors for breast cancer. The results showed that women aged ≥ 50 were more likely to use chemoprevention than women younger than 50 (28% versus 11%, *p* < 0.001). Moreover, the presence of risk factors for breast cancer increased chemoprevention uptake only in women aged ≥ 50 [43].

Whether improving the safety profile of chemoprevention might increase the uptake of preventive agents and consequently decrease breast cancer mortality is unknown.

Studies with low-dose tamoxifen have shown lower toxicity than and similar efficacy to higher doses [44]. A study randomized 500 women with intraepithelial neoplasia (atypical hyperplasia, LCIS or DCIS) to receive low-dose tamoxifen (5 mg/day) or placebo for three years [44]. The low-dose tamoxifen group showed half of the neoplastic breast events (DCIS or invasive cancer) that the placebo group did after a median follow-up of five years. Additionally, these results were consistent with the effect of 20 mg/day of the NSABP-B24 subgroup analysis of hormone-sensitive DCIS (HR 0.58; 95% CI 0.24–0.81); patient adherence to the treatment was similar in both groups [45].

At a median follow-up of 9.7 years, patients assigned to low-dose tamoxifen had a significant 42% reduction in neoplastic breast events (in situ or invasive); the annual rate per 1000 person-years was 11.3 for patients with tamoxifen versus 19.5 with placebo (HR 0.58, 95% CI 0.35–0.95; log-rank *p* = 0.03). In addition, the incidence of contralateral breast cancer was decreased by 64% for patients with tamoxifen (HR 0.36; 95% CI 0.14–0.92; *p* = 0.025). The number needed to be treated with tamoxifen to prevent one case of a breast event was 22 in five years and 14 in ten years. Low-dose tamoxifen reduced recurrence by 50% (HR 0.50; 95% CI 0.28–0.91; *p* = 0.02) in the DCIS cohort, the subgroup representing 70% of the overall population. Low-dose tamoxifen did not increase the risk of serious adverse effects, including deep venous thrombosis and endometrial cancer. Therefore, low-dose tamoxifen represents an alternative for women diagnosed with intraepithelial neoplasia [46].

Current agents prescribed for chemoprevention decreased breast cancer diagnoses, primarily the incidence of ER+ breast cancers. This selective benefit might be because the available agents target the hormonal pathways, while other factors trigger the progression of ER-negative breast cancer. Moreover, triple-negative breast cancers are more aggressive and have inferior survival than ER-positive tumors [47]. Chemoprevention did not decrease breast cancer-related mortality [17]. However, different from the screening programs, mortality is not the primary goal of chemoprevention, while decreasing breast cancer incidence may avoid a cancer diagnosis and aggressive therapies, besides reducing healthcare costs [11,39].

The E3N cohort assessed the association between breast cancer risk and low-dose aspirin or clopidogrel use in postmenopausal women [48]. Among 62,512 women followed during nine years, the authors identified 2864 breast cancer cases. A transient higher breast cancer risk was observed during the third year of low-dose aspirin use compared with never use (HR 1.49, 1.08–2.07), followed by a lower risk (HR 0.72, 0.52–0.99). Clopidogrel ever use was associated with a higher breast cancer risk (HR 1.3, 1.02–1.68), restricted to ER- tumors (3.07, 1.64–5.76, *p* = 0.01). The authors concluded that antiplatelet drugs are not good pharmacologic candidates for breast cancer prevention [48].

Metformin is an oral glucose-lowering agent used in first-line therapy for type 2 diabetes mellitus [49]. A systematic review and meta-analysis selected 11 independent studies to evaluate the impact of metformin on cancer incidence and mortality. The study reported 4042 cancer events and 529 cancer deaths in patients with diabetes. Patients using metformin had the relative risk reduced by 31% (95% CI 0.61–0.79) compared to other antidiabetic drugs. This inverse relation was notable for pancreatic and hepatocellular cancer but not for colon, breast, and prostate cancer [50,51,52]. This observation led to further investigations in primary breast cancer patients as prevention. According to this, the NCT01905046 trial has been designed to evaluate the role of metformin hydrochloride in reducing breast cancer occurrence in patients with atypical hyperplasia or in situ breast cancer [53].

Veronesi et al. evaluated the efficacy of the retinoic acid derivative fenretinide in reducing second primary breast cancers [54]. The study randomized 2972 patients with surgically removed breast cancer to receive fenretinide 200 mg/day for five years or a placebo. Results showed no benefit in preventing second primary breast cancer [54]. At a median follow-up of 14.6 years, a subgroup analysis showed a decreased risk of second breast cancer only in premenopausal women (HR 0.62; 95% CI 0.46–0.83) [55].

A meta-analysis assessed the effect of vitamin D supplementation in reducing breast cancer risk in 19,137 females. The analysis described no effect on breast cancer risk reduction (RR 1.04: 95% CI 0.84–1.28, *p* = 0.71) [56].

Two randomized clinical trials assessed the efficacy of alendronate and zoledronic acid in breast cancer risk reduction [57]. The Fracture Intervention Trial (FIT) randomized 6459 women aged between 55 and 81 years to receive alendronate or a placebo, with a mean follow-up of 3.8 years. The HORIZON-PFT (The Health Outcomes and Reduced Incidence with Zoledronic Acid Once Yearly-Pivotal Fracture Trial) randomly assigned 7765 women between 64 and 89 years old to receive annual intravenous zoledronic acid or a placebo for a mean follow-up of 2.8 years. Notwithstanding, neither alendronate nor zoledronic acid decreased the risk of postmenopausal breast cancer [57].

Preclinical studies assessed the role of various natural compounds in preventing breast cancer, including curcumin [58], genistein [59], resveratrol [60], and epigallocatechin gallate (EGCG) [61]. In vitro studies have shown that the flavonoid quercetin may enhance tamoxifen-induced antiproliferative effects [62]. However, further clinical studies are necessary to address the safety and efficacy of these compounds in breast cancer prevention, isolated or combined with other agents. 

### 2.2. Chemoprevention for Hereditary Breast Cancer

For women carrying BRCA1 and BRCA2 P/LP germline variants, the cumulative breast cancer risks to age 80 are estimated at 72% and 69%, respectively [63]. The gold standard for primary breast cancer prevention remains bilateral mastectomy, and the annual screening with magnetic resonance imaging and mammography enables earlier detection [64,65]. Data on the efficacy of tamoxifen, raloxifene, and aromatase inhibitors on breast cancer primary prevention in women carrying BRCA1 or BRCA1 and BRCA2 P/LP germline variants are scarce. 

The first evidence of breast cancer risk reduction with tamoxifen in healthy BRCA1 and BRCA2 germline variant carriers came from a subgroup analysis of the P-1 trial. The P-1 study evaluated the efficacy of tamoxifen (versus placebo) for reducing breast cancer incidence in 13,388 women at increased risk for breast cancer. Of 288 patients who developed breast cancer after being enrolled in the study, 19 (6.6%) carried BRCA1 P/LP (n = 8) or BRCA2 P/LP (n = 11) germline variants [66]. Tamoxifen did not decrease breast cancer incidence among healthy patients with BRCA1 germline variants. Of eight patients with BRCA1 germline variants who developed breast cancer, five had tamoxifen, and three received a placebo (RR 1.67; 95% CI, 0.32–10.7). Regarding BRCA2 germline variant carriers, of 11 patients with breast cancer, 3 received tamoxifen and 8 had a placebo (RR 0.38; 95% CI, 0.06–1.56). The subgroup analysis of the P-1 trial that assessed chemoprevention with tamoxifen in BRCA1 and BRCA2 germline variant carriers has limitations regarding the small number of patients. If the number of patients with BRCA2 germline variant was higher, the observed risk ratio could be statistically significant. In addition, the study was not designed to address tamoxifen chemoprevention specifically in BRCA1 and BRCA2 germline variant carriers. Again, it is unclear if prevention started before 35 years of age in patients with BRCA1 germline variants could have different results. The role of prophylactic bilateral salpingo-oophorectomy in breast cancer reduction is known and is most evident in younger women [67]. However, it is unknown if tamoxifen enhances this benefit. In addition, these findings may be related to the greater likelihood of developing ER+ tumors in BRCA2 germline variant carriers compared with BRCA1 germline variant carriers.

Although the evidence for chemoprevention with tamoxifen for primary breast cancer in BRCA germline variant carriers is controversial, studies have shown that tamoxifen reduces the occurrence of contralateral breast cancer [64]. According to a meta-analysis, treatment with tamoxifen for a first breast cancer reduced the risk of a second breast cancer in BRCA1 and BRCA2 germline variant carriers by 44% (HR 0.56; 95% CI 0.41–0.76), 0.47 (95% CI 0.37–0.60) for BRCA1 and 0.39 (95% CI 0.28–0.54) for BRCA2 germline variant carriers [68].

The randomized, double-blinded, placebo-controlled phase III French Liber Trial evaluated chemoprevention with aromatase inhibitors in postmenopausal women carrying BRCA1 or BRCA2 P/LP germline variants. The study compared the treatment with letrozole 2.5 mg/day for five years (n = 84) versus a placebo (n = 86) in decreasing breast cancer incidence [69]. The study population comprised postmenopausal women aged between 40 and 70, healthy or with unilateral breast cancer diagnosed five or more years earlier. After a median follow-up of 72.7 months, the 5-year invasive breast cancer-free survival did not differ between the two groups (92% for placebo and 91% for letrozole; HR 0.83; 95% CI 0.3–2.3, *p* = 0.73) in the overall population. Similar results were described in women with or without breast cancer and BRCA1 or BRCA2 carriers. Limitations included the small number of patients (170 of 270 expected) and the high dropout rate. 

The uptake of chemoprevention agents is low among women carrying BRCA1 and BRCA2 P/LP germline variants. Metcalfe et al. examined differences in the uptake of preventive practices (screening with mammography and MRI, prophylactic mastectomy, prophylactic oophorectomy, and chemoprevention with tamoxifen) by 2677 women with BRCA1 and BRCA2 P/LP germline variants from nine countries. Approximately half of the women at risk for breast cancer did not opt for preventive measures and relied solely on regular screening. On the other hand, 1531 (57.2%) women opted to undergo a bilateral prophylactic oophorectomy. Among the 1383 women who did not have breast cancer, 248 (18%) underwent a prophylactic bilateral mastectomy. For those who did not choose to have a prophylactic mastectomy as a preventive option, only 76 women (5.5%) decided to take tamoxifen and 40 (2.9%) raloxifene for breast cancer risk reduction. The uptake of the different preventive options varied among different countries. Women from the US were the most likely to take tamoxifen or raloxifene (12.4%), while no women from Norway, Italy, Netherlands, or France reported using these drugs. Furthermore, among women without breast cancer, those who had undergone an oophorectomy had a higher tamoxifen usage rate (15.6%) compared with those who had not undergone a prophylactic oophorectomy (1.7%) [70]. Table 1 presents the main characteristics of the studies that assessed the benefit of chemoprevention for sporadic and hereditary breast cancer.

## 3. Lifestyle and Reproductive Factors

### 3.1. Lifestyle and Reproductive Factors for Sporadic Breast Cancer

The association between breast cancer and overweight/obesity, fat diet, low physical activity, alcohol intake, and hormone replacement therapy is well known [9]. A healthy lifestyle, such as increased physical activity and reduced alcohol intake, may prevent around 15–40% of breast cancers [9].

The Vitamins and Lifestyle (VITAL) cohort study examined the association between the incidence of invasive breast cancer and six recommendations of the WCRF/AICR cancer prevention program, focusing on body fatness, physical activity, foods that promote weight gain, plant-based foods, red and processed meats, and alcohol consumption over a follow-up of 6.7 years [72]. The study included 30,797 postmenopausal women aged between 50 and 76 years at baseline (2000–2002) without a history of breast cancer. Breast cancers (n = 899) were monitored through the Western Washington Surveillance, Epidemiology, and End Results database. Women who met at least five recommendations showed a reduction in breast cancer risk by 60% compared with those who met none (HR 0.40; 95% CI 0.25–0.65). In addition, the reduction in breast cancer risk observed for women meeting the recommendations related to body fatness, plant foods, and alcohol intake compared with no recommendations was 62% (HR 0.38; 95% CI 0.25–0.58) [72].

A systematic review and meta-analysis conducted in January 2020 evaluated the effect of 15 preventable factors on breast cancer risk [73]. The RRs (95% CI) of the factors associated with breast cancer were 1.07 (1.05–1.09) for cigarette smoking, 1.10 (1.07–1.12) for alcohol drinking, 1.18 (1.13–1.24) for overweight/obesity in postmenopausal women, 1.16 (1.03–1.31) for nulliparity, 1.37 (1.25–1.05) for late pregnancy, and 1.26 (1.20–1.32) for ever HRT use; sufficient physical activity and fruit/vegetable consumption were associated with a decreased risk for breast cancer (0.9, 0.86–0.95 and 0.87, 0.83–0.90, respectively) [73].

The French E3N populational-based cohort study analyzed the relationship between physical activity and breast cancer incidence between 1990 and 2002 among 90,509 women between 40 and 65 years of age. A linear decrease in the risk of breast cancer with increasing amounts of moderate (*p* trend < 0.01) and vigorous (*p* trend, 0.0001) recreational activities was observed. Women who reported five or more weekly hours of vigorous recreational activity had a lower risk of breast cancer (RR 0.62; 0.49–0.78) than women who reported neither moderate nor vigorous recreational activity. In addition, physical activity remained protective for women at high risk of breast cancer based on BMI, family history of breast cancer, nulliparity, and HRT use [74].

The Women’s Contraceptive and Reproductive Experiences Study was a multicenter population-based case-control trial that included women aged between 35 and 64 with newly diagnosed invasive breast cancer. The study’s results showed that individuals with exercise activity levels exceeding the median activity level of the active control subjects had an approximately 20% lower risk of developing breast cancer (OR 0.82; 95% CI 0.71–0.93) compared with inactive subjects [75].

A prospective assessment examined the association between physical activity and breast cancer risk in 45,631 women from the US Radiologic Technologists cohort. The authors observed that women practicing walking/hiking ≥ 10 h per week had the most substantial risk reduction than those reporting no walking/hiking (RR 0.57; 95% CI 0.34–0.95) [76].

On the other hand, the relationship between body mass index (BMI) and breast cancer is more complex. Age and menopausal status modify the effect of BMI on the development of breast cancer [77]. Women with a high BMI are associated with a lower risk for premenopausal cancer, while overweight status after menopause increases the risk for menopausal cancer [77,78].

The prospective cohort Nurses’ Health study assessed the impact of weight change on the occurrence of invasive breast cancer in 87,143 postmenopausal women aged 30 to 55 years. The researchers followed women for 26 years, from age 18 (from 1976 to 2022). The study also evaluated weight change since menopause among 49,514 women over 24 years. The results revealed that women who experienced a weight gain of 25 kg or more since age 18 had a higher risk of developing breast cancer compared with those who had maintained their weight (RR 1.45; 95% CI 1.27–1.66, *p* < 0.01). In addition, women who gained 10 kg or more since menopause compared with weight maintenance had an increased risk for breast cancer (RR 1.18; 95% CI 1.0–1.35; *p* < 0.002) [79].

The PREDIMED study randomized 4282 women between 60 and 80 years old, from 2003 to 2009, at high cardiovascular risk to three different diets: a Mediterranean diet supplemented with extra-virgin olive oil, a Mediterranean diet supplemented with mixed nuts, or a control diet (advice to reduce dietary fat). After a median follow-up of 4.8 years, the Mediterranean diet nuts group showed a nonsignificant risk reduction compared with the control group (HR, 0.63; 95% CI 0.28–1.41). When both Mediterranean diet groups were merged, the risk relative reduced by 51% (95% CI, 0.24–0.98) [80].

Population-based studies have suggested that the impact of dietary composition on breast cancer risk might be particularly significant during adolescence and early adulthood [81]. A prospective cohort study assessed the association between fruit and vegetable intake during adolescence and early adulthood and the subsequent risk of breast cancer in a group of health professionals in the US [82]. The analysis included 90,476 premenopausal women aged between 27 and 44, from the Nurses’ Health Study II. These participants completed a questionnaire on diet in 1991. Additionally, 44,223 of them also provided information about their diet during adolescence in 1998. The results showed that higher total fruit consumption during adolescence was associated with a lower risk of breast cancer. Furthermore, the association remained independent of fruit intake during adulthood. Specifically, the HR was 0.75 (95% CI 0.62–0.90) for the highest (median intake 2.9 serving/day) versus the lowest (median intake 0.5 serving/day) fifth of intake [82].

A systematic review of 14 cohorts and 18 case-control studies assessed the associations between different dietary patterns and the risk of breast cancer [83]. The results described distinct associations between dietary patterns and breast cancer risk. A Western dietary pattern increased the risk of breast cancer by 14% (RR 1.14; 95% CI 1.02–1.28), while a healthy dietary pattern reduced the risk by 18% (RR 0.82, 95% CI 0.75–0.89). Subgroup analysis indicated that the positive association between a Western dietary pattern and breast cancer risk was significant among postmenopausal (RR 1.20; 95% CI 1.06–1.35) but not premenopausal women (RR 1.18; 95% CI 0.99–1.40). Furthermore, the study found that a Western dietary pattern was positively associated with hormone receptor-positive breast tumors (RR 1.18; 95% CI 1.04–1.33) but not with receptor-negative tumors (RR 0.97; 95% CI 0.83–1.12) [83].

The WHI DM trial randomized 48,835 postmenopausal women with no prior breast cancer, aged between 50 and 79, and consuming ≥ 32% of their energy from dietary fat to receive the usual diet comparison group (60%) or the dietary intervention group (40%). The goals were to decrease fat consumption by 20% and increase fruit, vegetable, and grain intake. After a median 19.6-year follow-up, the researchers described a noteworthy reduction in overall mortality after breast cancer (HR 0.85; 95% CI 0.74–0.96, *p* = 0.01) and a decrease in breast cancer-related deaths (HR 0.79; 95% CI 0.64–0.96, *p* = 0.02). There was also a significant reduction in worse-prognosis ER+ PR-negative (PR-) breast cancer (*p* = 0.01) occurrence in the low-fat dietary intervention group [84].

Another analysis evaluated 10 cohort studies, including 993,466 women followed for 11 to 20 years, documenting 19,869 ER+ and 4821 ER- breast cancers [85]. They reported a statistically significant inverse association between vegetable consumption and ER- breast cancer risk (HR 0.82; 95% CI 0.74–0.90) but not for breast cancer overall or ER+ tumors [85].

A meta-analysis including 572 studies and 486,538 cancer cases assessed the effect of alcohol on 23 cancer types. The results showed an increased risk for female breast cancer with an RR of 1.04 (95% CI 1.01–1.07) for light (≤12.5 g/day of alcohol), RR of 1.23 (95% CI 1.19–1.28) for moderate (≤50 and >50 g/day), and 1.61 (95% CI 1.33–1.94) for heavy drinking (>50 g/day) [86].

Another meta-analysis conducted a dose-response assessment between various alcohols and the risk of breast cancer. The analysis included 22 cohort studies involving 45,350 cases of breast cancer. The results showed a higher risk of developing ER+ tumors for current drinkers than for never-drinkers. The dose-response analysis indicated a significant linear correlation between breast cancer risk and total alcohol intake and wine consumption. For every additional 10 g of alcohol consumed per day, the risk of breast cancer increased by 10.5% (RR = 1.10; 95% CI 1.08–1.13) for total alcohol and 8.9% (RR = 1.08; 95% CI 1.04–1.14) for wine. In postmenopausal women, the risk of breast cancer increased by 11.1% (95% CI 1.09–1.13) with every 10 g increase in total alcohol consumption. Moreover, the study found that the percentage of breast cancer cases attributed to alcohol consumption was higher in Europe compared to North America and Asia [87].

Two nested case-control studies assessed the risks of breast cancer associated with different types and durations of HRT in the UK. The analyses included 98,611 women (59–79 years old) with a primary diagnosis of breast cancer between 1998 and 2018, matched by age, general practice, and index date to 457,498 female controls [88]. Overall, 33,703 (34%) women with a diagnosis of breast cancer and 134,391 (31%) controls had used HRT prior to one year before the index date. Compared with never use, in recent users (<5 years) with long-term use (≥5 years), estrogen-only therapy and estrogen and progesterone combined therapy were associated with increased risk for breast cancer (adjusted OD 1.15, 95% CI 1.98–1.21 and 1.79, 95% CI 1.73–1.85, respectively). For long-term (< of 5 years) HRT cessation, former estrogen-only HRT was no longer associated with BC increased risk. The risk was still present for combined estrogen and progestogen treatment (OD 1.16, 95% CI 1.11–1.21) [88].

A recent meta-analysis of individual participant data from prospective studies (from 1 January 1992 to 1 January 2018) included 108,647 postmenopausal women who developed breast cancer, of whom 55,575 (51%) had used HRT. Of interest, every HRT type, except vaginal estrogens, was associated with breast cancer risk, which steadily increased with the duration of use. In addition, the risk was higher for estrogen and progesterone than for estrogen-only preparations. The analysis included current users up to 5 years after last-reported HRT use. The participants had a clear excess risk even during the first four years (estrogen-progesterone RR 1.60; 95% CI 1.52–1.69; estrogen-only RR 1.17, 1.10–1.26); the risk increased to twice during years 5–14 (estrogen-progesterone RR 2.08, 2.02–2.15, estrogen-only RR 1.33, 1.28–1.37) [89].

Another meta-analysis assessed the association between breast cancer risk and parity, age at first birth, and breastfeeding. The analysis included 21,941 patients with breast cancer and 864,177 controls. The findings revealed that parity reduced the risk of the luminal subtype by 25% (OR 0.75; 95% CI, 0.70–0.81; *p* < 0.0001). Ever breastfeeding decreased breast cancer risk for both luminal and triple-negative subtypes (OR 0.77; 95% CI 0.66–0.88; *p* = 0.003 and OR 0.79; 95% CI 0.66–0.94; *p* = 0.01, respectively) [90].

Finally, the Nurses’ Health Study (1976–2012) and NHSII (1989–2013) investigated the association between breastfeeding, parity, and breast cancer risk, considering hormone receptor and molecular subtypes. A total of 12,452 (ER+ n = 8235, ER− n = 1978) breast cancer cases were diagnosed among 199,514 women. They observed that parous women compared with nulliparous women had a lower risk of ER+ breast cancer (HR 0.82; 0.77–0.88). In addition, among parous women, breastfeeding was associated with a lower risk of ER- versus never breastfeeding (HR 0.82; 0.74–0.91) [91].

### 3.2. Lifestyle and Reproductive Factors for Hereditary Breast Cancer

Besides intensified surveillance, chemoprevention, and risk reduction surgeries, strategies for decreasing breast cancer risk for P/LP germline variant carriers also comprise lifestyle factors [92].

A systematic literature review investigated whether physical activity levels during adolescence and young adulthood could decrease the lifetime risk of breast cancer among individuals carrying BRCA1 or BRCA2 germline variants. The review identified five relevant articles that met inclusion criteria and utilized self-reported physical activity data during these development stages. Among these studies, one assessed sports involvement, while the others focused on recreational activities. Four studies reported decreased breast cancer incidence over a lifetime with physical activity levels during adolescence and young adulthood. However, one study reported limited protection for premenopausal breast cancer (OR 0.62; 95% CI 0.40–0.96, *p*-trend = 0.01). Moreover, another study revealed a connection between adolescent and young adult physical activity and older age at breast cancer diagnosis (*p* = 0.03). Based on the limited evidence available, there are indications that physical activity during the formative years of adolescence and young adulthood may decrease or postpone the occurrence of breast cancer in individuals carrying BRCA1 and BRCA2 P/LP germline variants [93].

Another systematic review evaluated the evidence of dietary habits, weight status/change, and physical activity on ovarian and breast cancer risk among women with BRCA1/BRCA2 P/LP germline variants. Analysis suggested that higher diet quality, losing 10 pounds during adulthood, and engaging in physical activity during adolescence and young adulthood might be associated with a reduced risk of breast cancer. On the other hand, higher meat and daily energy intake could increase breast cancer risk. However, they concluded that there is currently insufficient evidence to propose individualized recommendations for dietary habits or weight management specifically for women with BRCA1 and BRCA2 germline variants compared to the general population regarding breast cancer risk reduction. Therefore, they recommend that dietary and physical activity recommendations remain the same for all women [94].

Kostopoulos et al. evaluated the association between weight gain or loss and the risk of breast cancer in a matched case-control study on 1073 pairs of BRCA1 (n = 797) and BRCA2 (n = 276) P/LP germline variant carriers [95]. The results showed an association between a decrease of at least 10 pounds from age 18 to 30 and a reduced risk of breast cancer between age 30 and 49 (OR 0.47; 95% CI 0.28–0.49). However, weight gain during the same period did not affect the overall risk. Among the BRCA1 germline variant carrier subgroup with at least two children, gaining over 10 pounds between ages 18 and 30 increased the risk of breast cancer diagnosed between ages 30 and 40 (OR 1.44; 95% CI 1.01–2.04). Furthermore, changes in body weight later in life (between 30 and 40 years old) did not impact the risk of premenopausal or postmenopausal breast cancer [95].

Another meta-analysis included three cohort studies for 1100 healthy women with BRCA1 and BRCA2 P/LP germline variants who underwent risk-reducing bilateral salpingo-oophorectomy before the onset of natural menopause to assess the association between breast cancer incidence and HRT. The results did not find an increased risk associated with using HRT beyond the baseline increase in the risk of breast cancer for women carrying BRCA germline variants [96].

A matched case-control study included 1665 pairs of women with BRCA1 (n = 1243 pairs) and BRCA2 (n = 422 pairs) P/LP germline variants to assess the association between breastfeeding and breast cancer risk. They observed a 32% risk reduction (OR 0.68; 95% CI 0.52–0.91; *p* = 0.008) in breast cancer for breastfeeding for at least one year among BRCA1 germline variant carriers and a more significant decrease in risk for two or more years of breastfeeding (OR 0.51; 95% CI 0.35–0.74). However, results showed no significant association between breastfeeding for at least one year and breast cancer risk among BRCA2 germline variant carriers (OR 0.83; 95% CI 0.53–1.31; *p* = 0.43) [97].

Khincha et al. conducted the first study to assess the association between female reproductive factors and breast cancer risk in women carrying a germline TP53 P/LP variant [98]. The researchers collected questionnaire data on 152 women enrolled in the National Cancer Institute’s LFS study, of which 85 had breast cancer. They found an association between lifetime breastfeeding for at least 12 months and a decreased breast cancer risk (HR 0.49, 95% CI 0.26–0.89, *p* = 0.02). Women who had their first live birth after the age of 30 years had a slightly increased breast cancer risk (HR 2.14; 95% CI 0.99–4.6, *p* = 0.05). Parity (HR 1.08, *p* = 0.8), age at menarche (HR 1.09, *p* = 0.24), and use of oral contraceptives (HR 0.88, *p* = 0.7) did not independently change breast cancer risk [98].

## 4. Conclusions and Future Directions

Breast cancer is the most commonly diagnosed cancer in women. For this reason, lifestyle recommendations to decrease breast cancer risk, such as increased physical activity, healthy diet, and reduced alcohol consumption, should be counseled to everyone. Breastfeeding should also be encouraged. Implementing specific preventive strategies should always be discussed during counseling and personalized risk assessment. The choice among different chemoprevention agents should consider patient comorbidities and each agent’s adverse effect profile. In addition, effective programs to increase chemoprevention agent uptake should be implemented.

Our group is now evaluating multimodal and combination strategies. We are starting a randomized phase II trial for breast cancer prevention, including healthy high-risk women and patients with intraepithelial neoplasia. The intervention groups will be low-dose tamoxifen (10 mg every other day); low-dose tamoxifen combined with intermittent caloric restriction (ICR two days/week); lifestyle intervention (LSI diet according to WCRF recommendations and step counter); and LSI combined with ICR. We hypothesize that the combination of low-dose tamoxifen and intermittent caloric restriction will improve risk biomarkers with a better quality of life. Furthermore, we aim to peruse the low dose regiments, and we are designing a study for post-menopausal women with intraepithelial neoplasia comparing tamoxifen 10 mg every other day versus exemestane 25 mg every other day, balancing efficacy, side effects, and quality of life.

## Figures and Tables

**Table 1 healthcare-11-02360-t001:** Characteristics of trials evaluating chemoprevention in breast cancer.

Study Name Author, Year (Reference)	Study Design	N of Participants	Intervention Arm	Control Arm	DoT	Primary Endpoints	Median Follow Up	Findings
NSABP P-1 trial Fischer et al., 1998 [21]	RCT	13,388	Tamoxifen 20 mg/day	Placebo	5 years	Risk of occurrence of BC	5 years	1. Risk reduction of IBC by 49% (two-sided *p* < 0.00001) and of NIBC by 50% (two-sided *p* < 0.002) in the tamoxifen group 2. Higher risk of endometrial cancer in the tamoxifen group (RR 2.53; 95% CI 1.35–4.97)
Intervention Breast Cancer Intervention Study (IBIS-I trial) Cuzick et al., 2007 [24]	RCT	7145	Tamoxifen 20 mg/day	Placebo	5 years	Risk of occurrence of BC	96 months	1. Risk reduction effect of tamoxifen appears to persist for at least 10 years (RR 0.73; 95% CI 0.58–0.91, *p* = 0.004). 2. Most side effects of tamoxifen do not continue after the 5-year treatment period.
Royal Marsden Hospital study Powles et al., 2007 [27]	RCT	2471	Tamoxifen 20 mg/day	Placebo	up to 8 years	Risk of occurrence of BC	13 years	1. The risk of ER+ BC was not statistically significantly lower in the tamoxifen arm than in the placebo arm during the 8-year treatment period (HR 0.77; 95% CI 0.48–1.23, *p* = 0.3) but was statistically significantly lower in the post-treatment period (HR 0.48; 95% CI 0.29–0.79, *p* = 0.004).
Italian Tamoxifen Prevention Study Veronesi et al., 2007 [29]	RCT	5408	Tamoxifen 20 mg/day	Placebo	5 years	Occurrence of BC and deaths of BC	11 years	1. The rates of breast cancer in the two study groups were similar among women who had a low risk for HR+ BC but were much lower in the tamoxifen group among women at high risk (RR 0.24; 95% CI 0.10 to 0.59).
Continuing Outcomes Relevant to Evista (CORE) Trial Martino et al., 2004 [32]	RCT	4011	Raloxifene 60 mg/day	Placebo	8 years	Incidence of IBC	8 years	1. The 4-year incidences of IBC and ER+ IBC were reduced by 59% (HR 0.41; 95% CI 0.24–0.71) and 66% (HR 0.34; 95% CI 0.18–0.66), respectively, in the raloxifene group. 2. Higher risk of thromboembolism in the raloxifene group (RR 2.17; 95% CI 0.83–5.70).
Raloxifene Use for The Heart (RUTH) Barrett-Connor et al., 2006 [33]	RCT	10,101	Raloxifene 60 mg/day	Placebo	5 years	Incidence of coronary events and IBC	5.6 years	1. Raloxifene reduced the risk of IBC (HR 0.56; 95% CI 0.38–0.83) 2. Increased risk of fatal stroke and venous thromboembolism.
NSABP STAR trial (P-2) Vogel et al., 2006 [35]	RCT	19,747	Tamoxifen 20 mg/day	Raloxifene 60 mg/day	5 years	Incidence of IBC, endometrial cancer, NIBC, bone fractures, and VTE	5 years	1. Similar incidence of IBC in both groups (RR 1.02; 95% CI 0.82–1.28); fewer noninvasive BC in the tamoxifen group (not statistically significant) 2. Lower incidence of VTE and endometrial cancer in the raloxifene group.
NCIC Clinical Trials Group Mammary Prevention.3 (MAP.3) trial Goss et al., 2011 [17]	RCT	4560	Exemestane 25 mg/day	Placebo	5 years	Incidence of IBC	35 months	1. A 65% relative reduction in the annual incidence of IBC (HR 0.35; 95% CI 0.18–0.70, *p* = 0.002). 2. No significant differences between the two groups in terms of skeletal fractures or CVC events.
Intervention Breast Cancer Intervention Study (IBIS-II trial) Cuzick et al., 2020 [18]	RCT	3864	Anastrozole 1 mg/day	Placebo	5 years	Incidence of IBC and NIBC	131 months	1. A 49% reduction in BC was observed for anastrozole (HR 0.51; 95% CI 0.39–0.66, *p* < 0.0001). 2. A 54% reduction in ER+ IBC (HR 0.46; 95% CI 0.33–0.65, *p* < 0.0001) 3. A 59% reduction in NIBC (HR 0.41; 0.22–0.79, *p* = 0.0081).
TAM-01 Study Lazzeroni et al., 2023 [46]	RCT	500	Tamoxifen 5 mg/day	Placebo	3 years	Incidence of IBC or NIBC	9.7 years	1. There were 66 breast cancers: 25 in the tamoxifen group and 41 in the placebo group. Significant 42% reduction of recurrence with tamoxifen. 2. NNT: 22 in 5 years and 14 in 10 years. 3. Significant 50% reduction in the NIBC (70% of the overall population). 4. No difference in SAE incidence during the follow-up period.
Intervention Breast Cancer Intervention Study (IBIS-I trial) King et al., 2001 [66]	RCT? (cohort retrospective)	288	Tamoxifen 20 mg/day	Placebo	5 years	Risk of occurrence of BC	5.7 years	1. Of the 288 breast cancer cases, 19 (6.6%) inherited disease-predisposing BRCA1 or BRCA2 mutations. 2. Of 8 patients with BRCA1 mutations, 5 received tamoxifen and 3 received placebo (RR 1.67; 95% CI 0.32–10.70). 3. Of 11 patients with BRCA2 mutations, 3 received tamoxifen and 8 received placebo (RR 0.38; 95% CI 0.06–1.56).
French Liber Trial Singer et al., 2020 [71]	RCT	170	Letrozole 2.5 mg/day	Placebo	5 years	BC incidence in postmenopausal women with gBRCA1/2 mutations	72.7 months	1. The 5-year BC-free survival did not significantly differ between the arms (HR 0.83; 95%CI 0.3–2.3, *p* = 0.73) in the overall population.

RCT—randomized controlled trial, BC—breast cancer, IBC—invasive breast cancer, NIBC—noninvasive breast cancer, ER—estrogen receptor, RR—relative risk, HR—hazard ratio, CI—confidence interval, VTE—venous thromboembolism, CV—cardiovascular, NNT—number needed to treat.

## Data Availability

Not applicable.

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
