# Peer review of "Chemoprevention and Lifestyle Modifications for Risk Reduction in Sporadic and Hereditary Breast Cancer"

_healthcare, 2023, doi:10.3390/healthcare11162360_

Round 1

Reviewer 1 Report

The manuscript provides a comprehensive review of the efficacy of chemoprevention and lifestyle interventions in women with sporadic and hereditary breast cancer. The topic is highly relevant considering the global impact of female breast cancer and the potential for risk assessment and personalized interventions to reduce its incidence. Overall, the manuscript is well-structured and provides a clear overview of the current evidence in this field.

However, I feel the review has not covered all aspects of breast cancer chemoprevention. The review is about chemoprevention and lifestyle modifications but the authors have not discussed a major factor of lifestyle and that is dietary modifications. The authors have not at all discussed the role of natural/dietary chemopreventive agents against breast cancer chemoprevention. There have been several clinical trials with promising results to show the efficacy of natural dietary chemopreventive agents against breast cancer for eg. PMID: 19901561; PMID: 22332908; PMID: 23725149. These are just a few examples. There is a plethora of such studies. Authors should include a section about such studies and their outcomes and also how the combination of natural/dietary chemopreventive agents with common antineoplastic drugs (tamoxifen, raloxifene, etc.) can reduce the side effects of such antineoplastic drugs and how that helps the patients lead a normal low side effect or a side effect free life during cancer treatment.

The manuscript is generally well-written, but there are a few instances where sentence structure or word choice could be improved for clarity. It would be beneficial to carefully proofread the manuscript for any grammatical errors or typographical mistakes.

Author Response

Dear Reviewer,

Thank you for giving me the opportunity to submit a revised draft of my manuscript “Chemoprevention and lifestyle modifications for risk reduction in sporadic and hereditary breast cancer” to Healthcare.

We appreciate the time and effort that you and the reviewers have dedicated to providing your valuable feedback on our manuscript. We are grateful to the reviewers for their insightful comments on my paper.

We have been able to incorporate changes to reflect most of the suggestions provided by the reviewers. We have highlighted the changes within the manuscript.

Comments from reviewer 1:

Comment 1: Discussion of the role of natural/dietary agents against breast cancer chemoprevention.

Response: Thank you for pointing this out. It would have been interesting to explore this aspect. However, in the case of our study, it seems slightly out of scope because we focused on primary prevention. We did not include papers that evaluated the role of natural/dietary agents in the treatment of patients who have already developed breast cancer. Even so, we were able to include three paragraphs regarding the role of natural/dietary modifications in breast cancer primary prevention, alone and in combination with tamoxifen (chemoprevention). We have highlighted the changes within the manuscript and we added a short paragraph about the potential role of natural compounds such as curcumin, genistein, resveratrol, and epigallocatechin gallate.

For more details, please see the revised manuscript.

Reviewer 2 Report

This review summarized the recent research or analysis on preventing female breast cancer, especially through chemoprevention and lifestyle modifications. These studies include detailed investigation, analysis, and experiments on the general population and pathogenic/likely pathogenic (P/LP) germline variants. There are a few reviews with partially overlapped topics published in the past several years in this area, including DOI: 10.1038/s41568-020-0266-x, DOI: 10.3390/nu11071514 and more. However, the manuscript is still interesting and meaningful due to the detailed analysis and plenty of references. However, the structure of the manuscript should have been better organized and may need further adjustment.  Overall, the manuscript is suitable for publication in Healthcare, subject to the minor revisions described below.

1.       Line 28, it should be 684,996 rather than 684.996.

2.       Line 56, 57, 238 and 272. The chapter titles for these parts are quite confusing. I think “Risk-reduction interventions” is not suitable for Chapter 2 as this part only focus on chemoprevention. And based on my understanding, both 2.1 and 2.2 are chemoprevention for sporadic breast cancer while 2.3 is chemoprevention for hereditary breast cancer. I suggest merging the chapters 2.1 and 2.2 and changing the chapter 2.3 to 2.2 using the following titles:

2. Chemoprevention
2.1 Chemoprevention for sporadic breast cancer (including current parts 2.1 and 2.2)
2.2 Chemoprevention for hereditary breast cancer (current 2.3)

3.       Line 338, 480, 540. Now there are two Chapter 4. Chapter 3 and first Chapter 4 both focus on lifestyle and reproductive factors but for different populations so I suggest merging them. Second Chapter 4 should be kept an independent part. In this way, the title could be as the following:

3. Lifestyle and reproductive factors
3.1 Lifestyle and reproductive factors for sporadic breast cancer (current 3)
3.2 Lifestyle and reproductive factors for hereditary breast cancer (current first 4)
4. Conclusions (current second 4)

Author Response

Dear Reviewer,

Thank you for giving me the opportunity to submit a revised draft of my manuscript “Chemoprevention and lifestyle modifications for risk reduction in sporadic and hereditary breast cancer” to Healthcare.

We appreciate the time and effort that you and the reviewers have dedicated to providing your valuable feedback on our manuscript. We are grateful to the reviewers for their insightful comments on my paper.

We have been able to incorporate changes to reflect most of the suggestions provided by the reviewers. We have highlighted the changes within the manuscript.

Comment 1: The structure of the manuscript should have been better organized.

Response: Thank you for this suggestion. We agree with this and have incorporated your suggestion throughout the manuscript. We have highlighted the changes within the manuscript.

For more details, please see the revised manuscript.

Reviewer 3 Report

Chemoprevention of cancer is a very relevant theme for investigation, and it should be discussed with a balance between its benefits and risks, namely cumulative, long-, and late-term risks, as these drugs will be used for many years. Better evidence of these risks could increase the intake of chemoprevention. However, the objective of the present study is to summarize "... the most recent evidence regarding the efficacy of chemoprevention and lifestyle interventions in women with sporadic and hereditary breast cancer."

The authors aim to present the most recent evidence but no novelty seems to be extracted from the revision of the literature.

The authors present only results of published studies without any discussion of these or recommendations for future research, or identification of research gaps.

The report of results in the text could be complemented with a visual summary (figures, bar graphs, etc).

Author Response

Dear Reviewer,

Thank you for giving me the opportunity to submit a revised draft of my manuscript “Chemoprevention and lifestyle modifications for risk reduction in sporadic and hereditary breast cancer” to Healthcare.

We appreciate the time and effort that you and the reviewers have dedicated to providing your valuable feedback on our manuscript. We are grateful to the reviewers for their insightful comments on my paper.

We have been able to incorporate changes to reflect most of the suggestions provided by the reviewers. We have highlighted the changes within the manuscript.

Comments from reviewer 3:

Comment 1: The authors present only results of published studies without any discussion of these or recommendations for future research, or identification of research gaps.

Response: We have been able to incorporate changes to reflect most of the suggestions regarding recommendations for future research and the identification of research gaps.

Comment 2: The report of results in the text could be complemented with a visual summary (figures, bar graphs, etc).

Response: Thank you for pointing this out. However, we decided to include a table summarizing all relevant studies included in this review.

For more details, please se the revised manuscript.

Round 2

Reviewer 1 Report

The manuscript can be accepted in its present format 

Minor spacing errors. Can be corrected before the final version is published.

Author Response

Dear Reviewer,

Thank you for pointing this out. We have edited the text for clarity, conciseness, readability, and corrected the spacing errors. 

For more details, please see the revised manuscript.

Reviewer 3 Report

Dear authors,

I acknowledge the efforts to summarize the evidence on the efficacy of chemoprevention as it is the objective of the manuscript, but I consider that it is a biased perspective on this theme. An active search for presenting the acute, long, and late adverse effects of chemoprevention should have been included in the objectives of the study. 

Author Response

Dear Reviewer,

Thank you for pointing this out. We agree with this and have incorporated your suggestion (lines 195-203), followed by a discussion regarding adverse effects. Kind regards.

For more details, please see the revised manuscript.